# Can active learning techniques simultaneously develop students' hard and soft skills? Evidence from an international relations class

**Andrea Betti** [1]◉*, **Pablo Biderbost** [1]◉, **Aurora García Domonte** [2]◉

1 Department of International Relations, Universidad Pontificia Comillas, ICAI-ICADE, Madrid, Spain,
2 Department of Finance, Universidad Pontificia Comillas, ICAI-ICADE, Madrid, Spain

◉ These authors contributed equally to this work.
* abetti@comillas.edu

## Abstract

**Data Availability Statement:** All relevant data are within the manuscript and its Supporting Information file.

**Funding:** The Pontifical University of Comillas, ICAI-ICADE (Madrid) financed the research for this

### Purpose

In recent decades, educators have pushed to implementing active learning techniques that can advance students' competences. Universities are increasingly required to develop knowledge measured in terms of grades (hard skills) and inter-personal, social, and communication abilities (soft skills). Nevertheless, within the field of active learning, educators often focus on how these techniques can improve students' hard skills and their satisfaction. Few have analysed whether and how these techniques might improve students' soft skills. Moreover, among these few studies, the majority has analysed hard and soft skills separately, measuring whether different active learning techniques may or may not improve them. Virtually no one has studied whether students' hard and soft skills can converge or diverge in an active learning format. This study allows us to understand the relations between these two sets of variables, for example, whether an improvement (or deterioration) in the hard skills corresponds to an improvement (or deterioration) in the soft skills, and vice versa.

### Method

In our experiment, we analyse the impact of a specific active learning format, such as the Flipped Classroom (FC), on both students' hard and soft skills, by comparing it with a traditional class integrated with other active learning techniques, such as presentations, debates, and teamwork activities. First, we use Pearson correlations to measure the relation between students' hard skills, understood in terms of grades, and a set of soft skills, such as critical thinking, self-efficacy, teamwork, and perception of learning. Second, we use canonical correlations to analyse whether hard and soft skills converge or diverge in an FC format, in comparison with a traditional teaching format integrated with the other active learning techniques.

article (Name of the research project: "Teaching Innovation in International Relations: A Comparative Study of the Flipped Classroom and the Traditional Classroom"). The funders had no role in study design, data collection and analysis, decision to publish, or preparation of the manuscript.

**Competing interests:** Competing interests: The authors have declared that no competing interests exist.

## Results and conclusions

Our main finding is that the FC *per se* neither improves nor worsens students' performance in terms of hard and soft skills.

## Introduction

In recent decades, educators have pushed to implement active learning techniques that can advance students' competences [1, 2]. Universities are increasingly required to develop knowledge measured in terms of grades (hard skills) and inter-personal, social, and communication abilities (soft skills). Nevertheless, within the field of active learning, educators often focus on how these techniques can improve students' hard skills and their satisfaction. Few have analysed whether and how these techniques might improve students' soft skills.

Moreover, among these few studies, the majority has analysed hard and soft skills separately, measuring whether different active learning techniques may or may not improve them. Virtually no one has studied whether students' hard and soft skills can converge or diverge in an active learning format. This study allows us to understand the relations between these two sets of variables, for example, whether an improvement (or deterioration) in the hard skills corresponds to an improvement (or deterioration) in the soft skills, and vice versa. This is particularly important in a "knowledge-driven" society, which increasingly includes soft skills in the "employability" requirements of students. These refer to the capacity to "meet the needs of employers" in a highly competitive job market [3, 4].

We fill the gaps in this study. In our experiment, we analyse the impact of a specific active learning format, such as the Flipped Classroom (FC), on both students' hard and soft skills, by comparing it with a traditional class integrated with other active learning techniques, such as presentations, debates, and teamwork activities. First, we use Pearson correlations to measure the relation between students' hard skills, understood in terms of grades, and a set of soft skills, such as critical thinking, self-efficacy, teamwork, and perception of learning. Second, we use canonical correlations to analyse whether hard and soft skills converge or diverge in an FC format, in comparison with a traditional teaching format integrated with the other active learning techniques. Our main finding is that the FC *per se* neither improves nor worsens students' performance in terms of hard and soft skills. We use heliographs to visualise our findings.

The first section reviews the literature on the FC by highlighting its main gaps. The second describes the implementation of an FC in an International Relations (IR) class and the quantitative methodology used in the experiment. The third discusses the results and their implications for active learning literature. The final section highlights the limitations of the study and some venues for future research.

## Related works

Despite being relatively extensive, the literature on the FC presents some gaps. First, the research tends to be based on "local evidence and experiences", mostly coming from U.S. higher education. Moreover, "systematic evidence on the effectiveness of the approach. . .based on empirical data" [5] is insufficient, although the number of empirical studies has recently increased. Since there is still "little work investigating student learning outcomes objectively", there is a necessity to conduct "controlled experimental or quasi-experimental designs" [6]. Finally, empirical analyses mostly come from the field of natural sciences. They are

considerably less frequent in social and political sciences. These areas would greatly benefit from research in understanding the best ways to implement the FC format, depending on the subjects and disciplines [7]. In this sense, a quantitative study aimed at empirically measuring the impact of the FC on the hard and soft skills of a group of Spanish IR students could positively contribute to this growing field.

Second, most studies compared the FC with traditional teaching, based on frontal lectures, in which the professor illustrates the contents and students passively listen [8]. Comparing too different teaching formats can make it difficult to detect their effects on students' skills. Moreover, the growing worldwide criticism against traditional teaching can make these comparisons biased, leading to the obvious but not very valuable conclusion that active learning is better than passive learning. An increasing number of scholars has observed this problem [9–11] and considered it more productive to systematically compare different types of FCs, for example "holistic" FCs based on different synchronous and asynchronous classrooms [12], self-regulated FCs in which students monitor their own learning performance [13], or FCs integrated with cooperative learning techniques [14].

In a similar way, others proposed to compare the FC format with other active learning formats, such as online teaching [15, 16], video-based classes [17], simulations [18], lecture capture back-up [19], or peer-reviewing [20, 21]. Along these lines, this study proposes a comparison between a fully FC format and a semi-traditional class based on frontal lecturing and other active learning techniques.

The majority of studies observed a positive effect of the FC on students' achievement (hard skills), measured in terms of grades [22–24]. One study specifically detected that for questions in which students have to apply knowledge, the FC can improve the students' exam results [24]. Positive effects were also found in terms of motivation [25, 26], engagement [27], and satisfaction [28, 29], although a minority detected a students' preference for traditional teaching [30], or for only partially flipped formats [31].

It is useful to study the effects on students' hard skills and perceptions to understand the advantages and disadvantages of an FC format. However, it risks neglecting other crucial aspects of learning, like "higher order thinking cognitive skills" [32]. For this reason, this study also analyses the effects of an FC format on a set of soft skills, highly prioritised in active learning literature.

Existing studies have mostly analysed the impact of FC and other active learning formats on hard and soft skills separately. For this reason, our main goal is to examine the possible correlations between hard and soft skills in an FC, compared with a semi-traditional class integrated with other active learning techniques. This is an important task, considering that interpersonal and social skills are increasingly considered "critical for productive performance in today's workplace" [33].

## Soft skills

First, we study the impact of an FC format on the development of students' self-efficacy. Psychologists conceptualised self-efficacy as what people believe about their capacities [34]. In the field of active learning, the term has been referred to as "students' beliefs about whether they are able to show certain learning behaviour." It is conceived as a "self-regulatory construct" that can improve students' "motivational profiles" and make them "more willing to put forth effort in the training" [35]. Although this skill has not been studied in the field of IR, scholars from other disciplines detected some positive effects of the FCs on both students' hard skills, (grades) and self-efficacy [36], especially when FCs are integrated with other active learning strategies, such as peer assessment [21] or mechanisms to "monitor and evaluate their own

learning performance" [13]. Nevertheless, other scholars implemented an FC by using Massive Open Online Courses (MOOC)s and, while they observed an improvement in students' grades, they did not detect any effect on students' self-efficacy or on the capacity to regulate their own learning [37].

Second, active learning scholars are unanimous about the necessity to find strategies to improve students' teamwork. According to Salas et al., "team leadership" means "the ability to direct and coordinate the activities of other team members"; "mutual performance monitoring", that is "the ability to develop common understandings of the team environment and apply appropriate task strategies"; backup behaviour, that is "the ability to anticipate other team members' needs", for example, shifting workload and achieving balance among members "during high periods of workload or pressure"; and "adaptability", understood as the "ability to adjust strategies." Other elements include "team orientation", which is the "belief in the importance of team goals over individual members' goals", "mutual trust", and communication [38]. These aspects relate teamwork to another active learning technique, such as "collaborative learning", a "method where a group of learners work collaboratively or together to achieve common objectives and goals" [39].

In disciplines such as IR [40] or engineering [41], some found that the FC can favour teamwork and collaborative learning, while others detected that, students are "less positively inclined toward the group work. . .in the flipped-class sessions" [31]. While Lambach et al [40] and Jenkins [31] only examined students' perceptions, Baytiyeh and Naja [41] also found a slight, non-significant, improvement of students' grades. But these studies are based on comparisons between FC and traditional formats. Among the scholars who compared the FCs with other active learning techniques, some found that FCs can improve both students' grades and teamwork skills, providing they are integrated with strategies that can improve the students' capacity to interact among each other, such as cooperative learning techniques [14] or gamification approaches [42].

Third, active learning scholars agree on the importance of critical thinking. But it is not easy to find a common definition of the concept, since it involves a wide range of descriptors, such as analysis, problem-solving, decision-making, reflection on "one's own thinking", and the capacity to evaluate outcomes of thought and action [43]. Some scholars identified five stages of critical thinking, such as "problem identification, problem definition, exploration, applicability, and integration" [44]. For Kong, critical thinking is the capacity to "think reflectively and judge skilfully, so as to decide what information is reliable and what actions should be taken during reasoning and problem solving" [45]. In their classic study about learning, Anderson et al. conceptualised critical thinking as the capacity to evaluate "a proposed solution", or "the reasonableness of a hypothesis", or "which of two alternative methods is a more effective and efficient way of solving given problems" [46]. In terms of active learning, it is common to conceptualise critical thinking in relation with other sub-skills, such as reflecting, arguing, evaluating, inferring, and drawing conclusions and implications [43].

In IR, we could only find one study that explored the impact of an FC on the capacity to "synthesise, develop an independent opinion and formulate critical statements" [41]. However, it was not based on a systematic empirical analysis of the effects of an FC format on critical thinking. Other political scientists identified either positive [47] or null effects [48] of active learning on critical evaluation skills, but they focused, respectively, on "collaborative hypertexts" and "debates", and not on FCs. In other disciplines, more systematic studies compared FCs with other active learning formats and found that, when FCs are integrated with techniques, such as peer-reviewing [20], games [42], or simulations [49], they can improve the students' capacity to apply, analyse, evaluate, and synthesise knowledge and information. Nevertheless, except for Murillo-Zamorano et al [42], who detected the positive impact of the

FC on both grades (hard skills) and critical thinking, no other study has analysed the effects of an FC format on both hard and critical thinking skills.

Finally, we focus on students' perception of learning. This is the most frequently studied skill in relation to students' hard skills. Some political scientists observed an improvement in both the grades and the perception of learning of students exposed to the FC, although only through a comparison with traditional teaching [50]. Studies from other disciplines that compared FCs with traditional teaching formats found a consistency in terms of an improvement in both grades and perception of learning [51]. Nevertheless, another study found that an improvement in grades does not necessarily mean an improvement in students' satisfaction [52]. After implementing active learning techniques other than the FC, other scholars detected a similar contradictory tendency. Students exposed to active learning improved their grades, compared to students taught in a traditional way. However, their perception was that they were learning less because they took "the cognitive effort associated with active learning" as an indicator of "poorer learning" [53].

The same debate can be found in those studies that compared different types of FCs or the FCs with other active learning formats. Some observed that FCs can produce consistent better grades and improved perception of learning [54], especially when they are integrated with peer assessment [17], the use of interactive platforms as Moodle [55], simulations [49], or online teaching [15]. Others found that students exposed to an FC format obtained better grades than students exposed to other active learning technologies, such as "lecture capture back-up", but they also expressed less learning satisfaction, as they associated the FC with a higher workload [19].

These contradictory tendencies remind us how important it is to study the possible correlations between hard and soft skills. Virtually all the studies that we know of separately measured the impact of FCs and other active learning formats on either hard skills or soft skills. Systematic analyses of how hard and soft skills can converge or diverge in an active learning format do not exist, beyond a few exploratory studies that do not belong to the field of active learning [56].

## Materials and methods

### Participants and experiment

Sixty-three students made up the sample of the experiment. They were divided in two groups, based on the alphabetical order of their last names. The University which held the experiment usually applies this criterion to divide the largest groups of students into subgroups. The experimental group was taught through a fully flipped format, while the control group was taught through a traditional class integrated with other active learning techniques. The groups did not know about each other's different methodology, nor could they share information or teaching materials, since the members of each group could only access the materials within their own group. Moreover, students did not freely create the groups. In the University where the experiment took place and, in our class too, groups are usually created by teachers or by the University itself, depending on the different tasks or different subjects. In this sense, there was no contamination between the experimental and the control group.

### Ethics statement

The pedagogical, logistical, and ethical aspects of this article and of its related research project (Name of the research project: "Teaching Innovation in International Relations: A Comparative Study of the Flipped Classroom and the Traditional Classroom") have been approved by the Teaching Innovation Evaluation Committee of the Pontifical University of Comillas,

during the academic year 2018–19. This Committee was part of the Teaching Innovation Unit, which operated under the responsibility of the Vice-Chancellor of the Pontifical University of Comillas.

Students voluntarily accepted to participate in the research. With this goal, they responded to a series of surveys that were administered via Moodle during the class Latin-American Studies, taught in the Dual Degree in International Relations and Business, during the academic year 2018–19. Students electronically consented and accepted to participate in the research. They entered the Moodle and provided their personal information. In case they refused to participate, they did not respond to the surveys. No minors were involved in the project.

**Class.**   Latin American Studies. We conducted the experiment in one specific block of this class, called "Politics in Latin America", which represented 33% of all classes–five weeks of one semester–and focused on the study of Latin American political systems. We conducted it on a limited portion of the class to forestall a disruptive implementation of the FC. Performing the experiment during the whole semester could create biases in the perceptions of students that are mostly used to traditional teaching.

**Degree.**   Dual Degree in International Relations (IR) and Business Administration (BA).

**Credits.**   6 ECTS.

**Type of class.**   Mandatory, core class.

**Language of teaching.**   English.

**Teacher.**   We aimed to study only the differences between the two methodologies (flipped vs. semi-traditional), so the professor was the same for both groups.

**Control group.**   Thirty-two students taught through a semi-traditional teaching format. The teacher taught two in-class hours per week through frontal lectures, where he explained the contents. The other two in-class hours per week were dedicated to a variety of active learning activities, such as teamwork, presentations, and debates, performed by using different types of resources (see below). We called this group zero "0".

**Experimental group.**   Thirty-one students taught through a fully flipped format. Following the most common definitions of the FC [57, 58], the teacher substituted frontal lectures with video-classes he pre-recorded and shared to students before the in-class hours. This way, the four in-class hours per week were fully dedicated to active learning activities. We called this group one "1".

The teacher used the same active learning techniques, both in the two hours per week dedicated to active learning in the control group, and in the four hours per week dedicated to the same goal in the experimental group. These are some of the applications used to perform these activities:

*Kaltura*. Free online video platform that allows both professors and students to pre-record videos that can be shared as teacher's lectures or students' presentations, by either including them in *PowerPoint* slides or by uploading them onto *Moodle*. The teacher used *Kaltura* to pre-record the video-lectures and send them to the students of the experimental group before the in-class hours.

*PowerPoint*. Presentation program used by the teacher to illustrate and explain the main contents of the frontal classes in the control group, and by students of both groups to explain the results of their research tasks during presentations and debates.

*Moodle*. Free and open-source learning management system that allows students and professors to share a variety of materials.

*Poll everywhere*. Free application available on the internet that allows teachers to perform instant polls among students, for example measuring their knowledge of specific contents. It was also used to create knowledge contests among teams of students.

The teacher used these techniques to perform the active learning activities in both the experimental and control group.

*Trivia quizzes.* It was used to create contests among students operating in teams. It allows the teacher to measure students' knowledge of the history and politics of Latin America by asking questions that teams must answer in real and limited time.

*Word search puzzles.* It was used to create contests among students operating in teams. Teams are required to identify political events, actors, and concepts related to Latin American history and politics through words placed in a grid.

*Photo find puzzles.* Technique that was used to create contests among students operating in teams. Teams are required to use images and pictures to identify political events, actors, and concepts related to Latin American history and politics.

*Case study.* The teacher assigned several case studies about the history and politics of Latin America that students had to research by working in groups. Students did the research by using **Google Scholar**, a freely accessible web search engine that allows them to browse many academic publications about specific subjects. This way, students delivered detailed analyses of specific political events, leaders, and phenomena. These analyses were later shared with the rest of the students and with the teacher through class presentations and debates. It allowed students to deepen their knowledge about actors and events studied in the class. Moreover, it helped them become familiar with qualitative techniques, such as discourse and document analysis [59].

**Debates.** Students were split into competing teams that had to debate specific topics. The teacher selected the topics and they all related to Latin America history and politics. For each topic, the teacher usually selected two opposing views, and each team was required to defend one perspective. During debates, teams had to "conduct a thorough analysis of the debate topics and prepare arguments in defence of their own positions and in refutation of those of their opponent" [51]. Debates are frequently used in social and political sciences because they get "students to engage multiple sides of an argument rigorously and consider new perspectives" [60]. Students are "required to make a specific choice and then develop an argument in support of that selection." Moreover, they are compelled to "approach a given issue from different perspectives by examining counterarguments" [64].

**Presentations.** The teacher required students to split up into groups and share their ideas and arguments about the case studies that they researched. Presentations usually lasted 20 minutes and allowed students to interact with the teacher and student audience. Members of the audience could challenge the presenters' points through questions and counterarguments [61].

**Collaborative learning.** Technique in which "students work together in small teams to investigate a topic and produce a group product that they then share with the whole class" [8]. By working together, students "reach their learning goals through discussion and peer feedback" from teachers and students [14]. Most of the active learning exercises used this technique in class, with students working in competing teams.

## Data collection

The collection of data about students included the following variables: membership of the experimental or control group, students' grades in the Degree in IR and students' grades in the Degree in BA. We did not contemplate variables, such as age, number of years spent at university, or family income. These variables would not have discriminated against our sample, as all the students were the same age, had similar socio-economic status, and were all enrolled in the third year of their Dual Degree.

We delivered two ANOVA tests to detect any difference between the control group and the experimental group, in terms of the average GPA in both degrees. The results did not show any statistically significant difference in the students' academic achievement prior to our experiment (ANOVA IR GDA p = 0.847; ANOVA BA GDA p = 0.716). This means that the sample was homogenous in terms of academic achievement. Before the experiment, there was no statistically significant difference between the two groups. We compared our distribution (alphabetical order) with two other possible ways to divide students into groups, such as student identification number and a fully random distribution. We did not find any significant difference, with the two groups being homogenous.

**Hard skills.** To collect the data on the hard skills, we used the students' average grades, a survey administered at the beginning of the block "Politics in Latin America", a survey administered at the end of the block, the students' scores in the part of the final exam dedicated to the block, and the students' final grades of the class.

First, we administered a preliminary survey at the beginning of the block "Politics in Latin American" about the contents of the block. This survey did not have any impact on students' final evaluation. The goal was only one way of measuring their knowledge of Latin American politics prior to the class. We called this survey TEST1. The numerical scale goes from 0 (no correct answer) to 10 (all correct answers). The minimum grade to pass a class in the Spanish university system is 5. In this first phase, as an initial measure of the student's academic achievement prior to the class, we used the average grade of their Degrees in IR and in BA. During their first two years, students could only take core mandatory classes, which makes the average measure homogenous for the whole sample. We called these average grades, respectively, IIRR and BA. TEST1, IIRR, and BA are the three variables referred to the hard skills in the initial moment of the experiment.

Second, at the end of the block, we administered the same survey to assess students' knowledge of Latin American politics after the explanation of the contents and the participation in the in-class activities, but before the students' preparation for the final exam. Since the result was not part of their class evaluation, students did not see this survey as an "exam," but, rather, as a control measure of their knowledge of the contents taught through one or the other methodology. We administered the survey without prior notice, so that students took it without studying beforehand. We called this survey TEST2.

Third, students did an exam at the end of the class that included some specific questions about the content of the block "Latin American Politics." This exam represented 50% of the final grade for this class. This exam was announced in due time, with students having the time to properly prepare it. We called this group of questions QUES.

Finally, as a measure of the students' general achievement, we included their final grade of the class, which we called FG.

The survey at the end of the block, the questions about the block in the final exam, and the final grade of the class are the three variables referred to the students' hard skills in the final moment of the experiment.

**Soft skills.** We collected data about students' skills in terms of teamwork, self-efficacy, critical thinking, and perception of learning, by administering a 25-question survey to both groups. Soft skills data was collected twice, at the beginning and at the end of the block, in order to assess the impact of the FC on students' soft skills. We called them SE (self-efficacy), TW (teamwork), LP (perception of learning) and CT (critical thinking).

For the questions about self-efficacy (one to ten) and perception of learning (seventeen), we used a Likert scale from zero to five, whereas zero is "completely disagree" and five "completely agree." For the questions about teamwork (eleven to sixteen), we used a scale to measure the perceptions of students, whereas zero is "very negative perception" and five "very positive

**Table 1. 25-question survey items about soft skills.**

| | Self-efficacy: SE | completely disagree | completely agree |
|---|---|---|---|
| 1 | I think I am going to get some excellent grades this year. | 0 | 5 |
| 2 | If I make an effort, I think I have enough capacity to achieve a good academic record. | 0 | 5 |
| 3 | I believe that I am able to understand even the most difficult topics in this course. | 0 | 5 |
| 4 | I think I have enough capacity to understand a subject, quickly and well. | 0 | 5 |
| 5 | I think I can pass the courses quite easily and even get good grades. | 0 | 5 |
| 6 | Although teachers are demanding and strict, I have great confidence in my own academic ability. | 0 | 5 |
| 7 | I think that I am prepared and well qualified to achieve academic success. | 0 | 5 |
| 8 | When they ask me to do projects or homework, I am sure that I will do them well. | 0 | 5 |
| 9 | I work effectively in any team, no matter who the teammates are. | 0 | 5 |
| 10 | Considering the difficulty of the degree, what I am learning, and my own abilities, I think I'll be fine when I finish (the degree). | 0 | 5 |
| | **Teamwork: TW** | **very negative perception** | **very positive perception** |
| 11 | Participation in teamwork sharing information, knowledge, and experiences. | 0 | 5 |
| 12 | Acceptance and compliance with the rules agreed upon in the group (deadlines, parts of the work, format, etc.). | 0 | 5 |
| 13 | Action to face the conflicts of the team in this subject. | 0 | 5 |
| 14 | Commitment to the management and operation of the team | 0 | 5 |
| 15 | Management of meetings effectively. | 0 | 5 |
| 16 | Communication and cohesion within the group. | 0 | 5 |
| | **Perception of learning: LP** | **few** | **a lot** |
| 17 | Regardless of your results in the exams, think how much you will learn in this section about Latin American politics | 0 | 5 |
| | **Critical thinking: CT** | **wrong** | **right** |
| 18–25 | Application of a set of questions about a fantasy story read by students in advance in order to test their capacities to use logical and critical thinking. | 0 | 1 |

perception." Finally, to measure critical thinking, we used questions (eighteen to twenty-five) with only one correct answer, so that the answers generated a dichotomous variable, whereas zero is "wrong" and one is "right." We used the average data for each soft skill (see Table 1).

Data was collected for the students of the control group (traditional teaching format integrated with other active learning techniques) and the experimental group (fully flipped teaching format). Data was collected by browsing the anonymized students' average grades, through the surveys administered in virtual form via Moodle, through the final exam administered in person, and by browsing the anonymized students' final grades of the class. Table 2 has the details of the data that we used and the names of the variables that we created.

## Data analysis

To analyse the data, we followed a two-step strategy. First, we performed Pearson correlations (PCs) between each hard and soft skill, for each group, and at each moment of the experiment: experimental group before the experiment, hard vs. soft; control group before the experiment, hard vs. soft; experimental group after the experiment, hard vs. soft; and control group after the experiment, hard vs. soft. This allowed us to detect the level of association among individual variables for each group, according to a before-after logic.

Second, we performed a Canonical Correlation Analysis (CCA), which allowed us to detect the level of association between both sets of variables, hard and soft skills, for each group, according to a before-after logic. CCA is useful to maximise the correlations among lineal combinations of variables [62]. Moreover, it simultaneously predicts the behaviour of multiple

**Table 2. Summary of variables.**

| DATA | MOMENT | TEACHING FORMAT | SKILL | VARIABLES |
|------|--------|-----------------|-------|-----------|
| Surveys | Before the experiment | Without FC | Hard | ContTEST1 |
| | | With FC | Hard | ExpTEST1 |
| | After the experiment | Without FC | Hard | ContTEST2 |
| | | With FC | Hard | ExpTEST2 |
| Average grade IIRR | Before the experiment | Without FC | Hard | ContIIRR |
| | | With FC | Hard | ExpIIRR |
| Average grade BA | Before the experiment | Without FC | Hard | ContBA |
| | | With FC | Hard | ExpBA |
| Grade of the block in the final exam | After the experiment | Without FC | Hard | ContQUES |
| | | With FC | Hard | ExpQUES |
| Final grade of the class | After the experiment | Without FC | Hard | ContFG |
| | | With FC | Hard | ExpFG |
| Self-efficacy | Before the experiment | Without FC | Soft | ContSE1 |
| | | With FC | Soft | ExpSE1 |
| | After the experiment | Without FC | Soft | ContSE2 |
| | | With FC | Soft | ExpSE2 |
| Teamwork | Before the experiment | Without FC | Soft | ContTW1 |
| | | With FC | Soft | ExpTW1 |
| | After the experiment | Without FC | Soft | ContTW2 |
| | | With FC | Soft | ExpTW2 |
| Perception of Learning | Before the experiment | Without FC | Soft | ContLP1 |
| | | With FC | Soft | ExpLP1 |
| | After the experiment | Without FC | Soft | ContLP2 |
| | | With FC | Soft | ExpLP2 |
| Critical Thinking | Before the experiment | Without FC | Soft | ContCT1 |
| | | With FC | Soft | ExpCT1 |
| | After the experiment | Without FC | Soft | ContCT2 |
| | | With FC | Soft | ExpCT2 |

variables. This helps make sense of the complexity of social reality, which is methodologically reflected in the multidimensional nature of concepts.

We used CCA in this study because the experiment complies with the requisites necessary for its implementation. On the one hand, there are, at least, ten observations for each variable. This way, it is possible to avoid the "data overfitting." On the other hand, data complied with the requirements of linearity, homoscedasticity, and multicollinearity [62].

The Canonical Correlation (CC) is usually represented this way:

$$Y1 + Y2 + Y3 + \cdots + Yn = X1 + X2 + X3 + \cdots + Xn \qquad (1)$$

The results below show that the CCA can be difficult to represent. Its outcome is not intuitive, meaning that it is hard to have a clear impression at a first glance. For this reason, Degani, Shafto & Olson [63] proposed using the heliograph to show the results of a CCA. The heliograph is a visualisation technique, represented as a sun conformed by concentric circles. Each of them represents different canonical functions. Those with greater magnitude are represented in the external circles. Those with lesser magnitude are represented in the internal circles (closer to the centre of the sun). Canonical functions, in this type of graph, can be related to one or more samples [64, 65].

**Table 3. Descriptive statistics of variables.**

|  | Obs. | Min | Max | Mean | Standard Deviation |
|---|---|---|---|---|---|
| ExpTEST1 | 31 | 2.000 | 6.000 | 4.306 | 0.928 |
| ExpIIRR | 31 | 6.790 | 9.270 | 7.707 | 0.522 |
| ExpBA | 31 | 6.410 | 9.240 | 7.363 | 0.636 |
| ExpSE1 | 31 | 1.600 | 4.800 | 3.842 | 0.857 |
| ExpTW1 | 31 | 2.670 | 4.830 | 3.941 | 0.553 |
| ExpLP1 | 31 | 2.000 | 5.000 | 4.194 | 0.833 |
| ExpCT1 | 31 | 0.000 | 0.750 | 0.463 | 0.198 |
| ContTEST1 | 32 | 2.000 | 7.000 | 4.391 | 1.293 |
| ContIIRR | 32 | 6.310 | 8.850 | 7.736 | 0.639 |
| ContBA | 32 | 5.720 | 8.810 | 7.296 | 0.811 |
| ContSE1 | 32 | 2.100 | 4.800 | 3.722 | 0.697 |
| ContTW1 | 32 | 2.830 | 4.830 | 3.765 | 0.500 |
| ContLP1 | 32 | 3.000 | 5.000 | 4.375 | 0.707 |
| ContCT1 | 32 | 0.250 | 0.880 | 0.585 | 0.151 |
| ExpTEST2 | 31 | 2.000 | 8.500 | 5.113 | 1.606 |
| ExpQUES | 31 | 4.380 | 10.000 | 8.006 | 2.036 |
| ExpFG | 31 | 6.750 | 9.780 | 8.513 | 0.774 |
| ExpSE2 | 31 | 1.500 | 5.000 | 4.016 | 0.800 |
| ExpTW2 | 31 | 2.500 | 5.000 | 3.887 | 0.581 |
| ExpLP2 | 31 | 2.000 | 5.000 | 4.226 | 0.805 |
| ExpCT2 | 31 | 0.250 | 0.750 | 0.555 | 0.151 |
| ContTEST2 | 32 | 2.500 | 7.500 | 4.969 | 1.350 |
| ContQUES | 32 | 1.880 | 10.000 | 8.204 | 1.955 |
| ContFG | 32 | 6.850 | 10.000 | 8.493 | 0.760 |
| ContSE2 | 32 | 1.200 | 5.000 | 3.809 | 0.780 |
| ContTW2 | 32 | 2.830 | 4.830 | 3.880 | 0.524 |
| ContLP2 | 32 | 3.000 | 5.000 | 4.406 | 0.665 |
| ContCT2 | 32 | 0.000 | 0.880 | 0.593 | 0.217 |

## Results

Table 3 contains the main descriptive statistics of the data that we used for our study.

Before the experiment, the average results of the variables referring to the hard skills, which reflected the students' academic achievement in both groups, were homogenous, with differences no higher than +/- 2%. Nevertheless, these differences were higher in the case of the soft skills, especially critical thinking, with a result that was 26% higher in the control group, compared to the experimental. At the end of the experiment, these differences between the two groups slightly varied; they were never higher than +/- 3% in the hard skills, and +/- 7% in the soft skills.

Table 4 contains the results of the PCs in the experimental group before the experiment. This table shows that teamwork was the only soft skill that significantly correlated with two of the three measurements of the hard skills (average Degree grades in IR and BA). This means that the students with the best average grades in the two Degrees were also the students who showed the largest development of this soft skill.

Table 5 contains the results of the CCs in the experimental group before the experiment. CCA produces various canonical (latent) functions. These represent, as Hair et al. [62] suggest, the correlational relationships between two linear composites (or canonical variates). The

**Table 4. Pearson correlations: Experimental group before the experiment.**

| Exp | | TEST1 | IIRR | BA | SE1 | TW1 | LP1 | CT1 |
|---|---|---|---|---|---|---|---|---|
| TEST1 | Pearson | 1 | | | | | | |
| | Sig. (bil) | | | | | | | |
| IIRR | Pearson | 0.17 | 1 | | | | | |
| | Sig. (bil) | (0.35) | | | | | | |
| BA | Pearson | 0.15 | **0.90** | 1 | | | | |
| | Sig. (bil) | (0.41) | **(0.00)**\*\*\* | | | | | |
| SE1 | Pearson | -0.03 | -0.04 | -0.06 | 1 | | | |
| | Sig. (bil) | (0.87) | (0.83) | (0.74) | | | | |
| TW1 | Pearson | -0.16 | **0.36** | **0.30** | **0.45** | 1 | | |
| | Sig. (bil) | (0.38) | **(0.05)**\*\* | **(0.10)**\* | **(0.01)**\*\*\* | | | |
| LP1 | Pearson | 0.07 | 0.14 | 0.06 | 0.03 | 0.26 | 1 | |
| | Sig. (bil) | (0.70) | (0.45) | (0.76) | (0.89) | (0.16) | | |
| CT1 | Pearson | -0.09 | -0.01 | 0.01 | -0.09 | 0.13 | 0.23 | 1 |
| | Sig. (bil) | (0.63) | (0.96) | (0.95) | (0.63) | (0.48) | (0.22) | |

\*\*\*. \*\*. \*: Significance difference at 1%, 5% and 10% level of confidence.

maximum number of canonical functions in a CCA is defined by the quantity of variables in the set with the fewest variables (in our case, three in the set that includes the hard skills). Following these authors, the canonical variates are the linear combinations that represent the weighted sum of the variables in each set. The relevant canonical functions are those with the highest magnitude of CC, statistical significance, and practical significance.

As Table 5 shows, there was no association between the behaviour of the variables referred to the soft skills and the behaviour of the variables referred to the hard skills. Students who obtained the best grades were not necessarily those who began with the best performance in the soft skills. No correlation had any statistical significance. This means that there was no reason to explore other measures related to the CCA.

Table 6 contains the results of the PCs in the control group before the experiment. It shows that self-efficacy was the only soft skill that significantly correlated with three measurements of the hard skills, that is the average grade of the Degree in IR, the average grade of the Degree in BA, and the initial survey. This means that the students with the best hard skills in terms of grades were also the students who performed best in this soft skill.

Table 7 contains the results of the CCs in the control group before the experiment. It shows no significant association between the behaviour of the variables referred to the soft skills and the behaviour of the variables referred to the hard skills. Students who obtained the best grades were not necessarily those who began with the best performance in the soft skills. Considering this result, there was no reason to explore other measures related to CCA.

Table 8 contains the results of the PCs in the experimental group after the experiment. It shows that perception of learning was higher for students who obtained the best grades in the

**Table 5. Canonical Correlations: Experimental group before the experiment.**

| | Correl. | Autovalor | Wilks St. | F | D.F. number | D.F. denom. | Sig. |
|---|---|---|---|---|---|---|---|
| 1 | 0.490 | 0.317 | 0.711 | 0.731 | 12.000 | 63.790 | 0.716 |
| 2 | 0.239 | 0.061 | 0.936 | 0.279 | 6.000 | 50.000 | 0.944 |
| 3 | 0.083 | 0.007 | 0.993 | | | | |

**Table 6.  Pearson correlations: Control group before the experiment.**

| Cont | | TEST1 | IIRR | BA | SE1 | TW1 | LP1 | CT1 |
|------|------|------|------|------|------|------|------|------|
| TEST1 | Pearson | 1 | | | | | | |
| | Sig. (bil) | | | | | | | |
| IIRR | Pearson | 0.28 | 1 | | | | | |
| | Sig. (bil) | (0.13) | | | | | | |
| BA | Pearson | 0.25 | **0.94** | 1 | | | | |
| | Sig. (bil) | (0.16) | **(0.00)*** | | | | | |
| SE1 | Pearson | **0.35** | **0.47** | **0.45** | 1 | | | |
| | Sig. (bil) | **(0.05)**** | **(0.01)*** | **(0.01)*** | | | | |
| TW1 | Pearson | 0.27 | 0.14 | 0.07 | **0.34** | 1 | | |
| | Sig. (bil) | (0.14) | (0.44) | (0.70) | **(0.06)*** | | | |
| LP1 | Pearson | 0.17 | 0.09 | 0.06 | 0.26 | -0.12 | 1 | |
| | Sig. (bil) | (0.35) | (0.64) | (0.73) | (0.14) | (0.50) | | |
| CT1 | Pearson | 0.02 | 0.23 | 0.25 | 0.24 | -0.19 | 0.01 | 1 |
| | Sig. (bil) | (0.90) | (0.21) | (0.16) | (0.20) | (0.30) | (0.97) | |

***. **. *: Significance difference at 1%. 5% and 10% level of confidence.

final survey. At the same time, the students who showed the best performance in critical thinking were also those who obtained the best final grades in the class. Moreover, unlike what happened before the experiment, teamwork did not significantly correlate with any of the measurements of the hard skills.

Table 9 contains the results of the CCs in the experimental group after the experiment. It shows that there was no significant association between the behaviour of the variables referred to the soft skills and the behaviour of the variables referred to the hard skills. Students who obtained the best grades were not necessarily those who best performed in the soft skills. Again, this means that there was no reason to explore other measures related to CCA.

Table 10 contains the results of the PCs in the control group after the experiment. It shows that self-efficacy was higher for students who obtained the best final grades. A similar pattern was visible before the experiment. This significant relation could be seen both in the final grade of the class and in the questions about the block "Politics in Latin America" in the final exam. Two other soft skills also had significant relations with the final grade of the class, perception of learning and critical thinking. These skills were higher for students who obtained the best final grades.

As Table 11 shows, there was only one canonical function with marginally statistical significance (0.063). Its magnitude was 67.7%. This type of measure should be considered acceptable when it passes 30% [64]. This is an indicator of the strength of the overall relationships among the linear composites (canonical variates) for the sets of variables. Moreover, it represents the bivariate correlation between the two canonical variates. The first column of Tables 12 and 13 show the canonical loadings related to each variable that composes each set.

**Table 7.  Canonical correlations: Control group before the experiment.**

| | Correl. | Autovalor | Wilks St. | F | D.F. number | D.F. denom. | Sig. |
|---|---|---|---|---|---|---|---|
| 1 | 0.535 | 0.400 | 0.641 | 1.014 | 12.000 | 66.435 | 0.446 |
| 2 | 0.317 | 0.111 | 0.897 | 0.482 | 6.000 | 52.000 | 0.818 |
| 3 | 0.051 | 0.003 | 0.997 | . | . | . | . |

**Table 8. Pearson correlations: Experimental group after the experiment.**

| Exp | | TEST2 | QUES | FG | SE2 | TW2 | LP2 | CT2 |
|---|---|---|---|---|---|---|---|---|
| TEST2 | Pearson | 1 | | | | | | |
| | Sig. (bil) | | | | | | | |
| QUES | Pearson | **0.33** | 1 | | | | | |
| | Sig. (bil) | **(0.07)*** | | | | | | |
| FG | Pearson | **0.40** | **0.68** | 1 | | | | |
| | Sig. (bil) | **(0.03)**** | **(0.00)***** | | | | | |
| SE2 | Pearson | 0.04 | 0.03 | 0.01 | 1 | | | |
| | Sig. (bil) | (0.84) | (0.86) | (0.96) | | | | |
| TW2 | Pearson | 0.15 | -0.07 | 0.18 | 0.21 | 1 | | |
| | Sig. (bil) | (0.42) | (0.69) | (0.34) | (0.26) | | | |
| LP2 | Pearson | **0.32** | -0.22 | 0.10 | 0.04 | 0.18 | 1 | |
| | Sig. (bil) | **(0.08)*** | (0.23) | (0.58) | (0.85) | (0.34) | | |
| CT2 | Pearson | 0.20 | 0.27 | **0.30** | 0.03 | -0.14 | 0.17 | 1 |
| | Sig. (bil) | (0.27) | (0.14) | **(0.10)*** | (0.85) | (0.46) | (0.36) | |

***. **. *: Significance difference at 1%. 5% and 10% level of confidence.

The canonical roots are the squared CCs, which provide an estimate of the amount of shared variance among the respective optimally weighted canonical variates. It is considered acceptable when it passes 10%. In this case, it had a value of 45.8% ($0.677^2$).

The redundancy index is the amount of variance in one set of variables. It is explained by the other canonical variate (and the other way around). The higher the canonical variate, the better. However, it has limited utility. CCA is not useful to maximise this result. In this case, the set composed by the hard skills explained 14.9% of the set composed by the soft skills. The other way around, the percentage increased to 19.9% (see Table 14).

- Tables eleven, twelve, thirteen, and fourteen provide typical examples of the difficulty to represent the results of CCA. For this reason, by using Microsoft Paint, we designed a heliograph that shows the results of the CCA in our experiment (see Fig 1). This allows to observe:

- Both sets of variables: set one and set two.

- The canonical function that provides four types of information: direction, intensity, statistical significance, and magnitude.

The direction can be observed in the colour and in the orientation of the bars: white bars built towards the outside represent positive relations, while black bars built towards the inside represent negative relations. The intensity can be observed in the size of the bars and in the numbers (canonical loadings). The bigger the bar, the higher the contribution of that variable

**Table 9. Canonical correlations: Experimental group after the experiment.**

| | Correl. | Autovalor | Wilks St. | F | D.F. number | D.F. denom. | Sig. |
|---|---|---|---|---|---|---|---|
| 1 | 0.580 | 0.506 | 0.571 | 1.256 | 12.000 | 63.790 | 0.267 |
| 2 | 0.357 | 0.146 | 0.859 | 0.657 | 6.000 | 50.000 | 0.685 |
| 3 | 0.125 | 0.016 | 0.984 | . | . | . | . |

**Table 10. Pearson correlations: Control group after the experiment.**

| Cont | | TEST2 | QUES | FG | SE2 | TW2 | LP2 | CT2 |
|---|---|---|---|---|---|---|---|---|
| TEST2 | Pearson | 1 | | | | | | |
| | Sig. (bil) | | | | | | | |
| QUES | Pearson | 011 | 1 | | | | | |
| | Sig. (bil) | (0.54) | | | | | | |
| FG | Pearson | 0.17 | **0.52** | 1 | | | | |
| | Sig. (bil) | (0.35) | **(0.00)**\*** | | | | | |
| SE2 | Pearson | -0.14 | **0.37** | **0.46** | 1 | | | |
| | Sig. (bil) | (0.46) | **(0.04)**\** | **(0.01)**\*** | | | | |
| TW2 | Pearson | -0.25 | 0.14 | 0.21 | 0.13 | 1 | | |
| | Sig. (bil) | (0.18) | (0.45) | (0.25) | (0.48) | | | |
| LP2 | Pearson | 0.02 | 0.24 | **0.42** | 0.09 | 0.13 | 1 | |
| | Sig. (bil) | (0.94) | (0.19) | **(0.02)**\** | (0.62) | (0.48) | | |
| CT2 | Pearson | 0.24 | 0.29 | **0.30** | 0.20 | -0.16 | 0.24 | 1 |
| | Sig. (bil) | (0.19) | (0.11) | **(0.09)**\* | (0.27) | (0.39) | (0.18) | |

\***. \**. \*: Significance difference at 1%. 5% and 10% level of confidence.

to the canonical function described. The statistical significance can be observed in the colour of the canonical loadings. The black ones describe a significant relationship, while the white describe a non-significant relationship. Finally, the magnitude of each canonical function is summarised in the lower end of the figure and is expressed by the RC values.

## Discussion

The results of our experiment do not show a clear pattern of convergence between soft and hard skills in either of the two groups. For this reason, it cannot be affirmed that the FC *per se* improves or worsens this relation. Even when they obtain a satisfactory academic achievement, students do not necessarily develop all the soft skills.

The results of the PCs in the experimental group before the experiment show that students with the best average grades in the two Degrees tended to develop teamwork the most (Table 4). Nevertheless, after applying the CCA, which allows us to simultaneously analyse the correlation between the soft skills and grades, we did not detect any significant relation (Table 5). Students who obtained the best grades were not necessarily those who performed best in terms of teamwork. The lack of convergence between hard skills and teamwork was even clearer after our experiment. The results of the PCs were not significant (Table 8). Moreover, the CCA shows that this skill did not significantly align with any of the measurements of the hard skills (Table 9). This is not in line with those studies that, after separately measuring the impact of an FC on grades and teamwork, detected a positive effect on both [14, 42].

**Table 11. Canonical correlations: Control group after the experiment.**

| | Correl. | Autovalor | Wilks St. | F | D.F. number | D.F. denom. | Sig. |
|---|---|---|---|---|---|---|---|
| 1 | **0.677** | 0.845 | 0.472 | 1.815 | 12.000 | 66.435 | **0.063**\* |
| 2 | 0.341 | 0.131 | 0.871 | 0.618 | 6.000 | 52.000 | 0.715 |
| 3 | 0.119 | 0.014 | 0.986 | . | . | . | . |

\*: Significance difference at 10% level of confidence.

**Table 12. Canonical correlation Set 1.**

| Variable | 1 | 2 | 3 |
|---|---|---|---|
| ContO2 | -.181 | -.966 | -.184 |
| ContPREG | .664 | -.364 | .654 |
| ContNF | .911 | -.294 | -.289 |

Instead, it could be in line with those who found that the FC can be detrimental to the development of teamwork skills [31].

Even though they did not conduct systematic analyses of the effects of the FC, other scholars identified, among its possible unintended consequences, the risk that the FC generates in students a sensation of abandonment and isolation [57]. While the FC can be useful to foster teamwork activities in the class, its general impact on students' habits can be different. Students mostly watch videos alone. This could favour an individualisation of their way of studying [66, 67]. This is certainly a positive outcome in terms of developing students' autonomy, but not necessarily in terms of learning how to work and collaborate with their classmates. Moreover, the fact that they cannot interact with the teacher or with other students during the recorded video-lectures could be harmful, especially for low-achieving students [68].

The results of the PC in the control group before the experiment show that there was a correlation between students' performance in terms of grades (hard skills) and self-efficacy (Table 6). This is the group that was taught through the traditional teaching format integrated with active learning techniques. Nevertheless, after performing the CCA, we did not detect any significant correlation between the performance of the students in the control group in terms of grades and their performance in any of the soft skills, including self-efficacy (Table 7). Moreover, after the experiment, the results of the CCA were not significant either (Table 11). Nevertheless, the PC of the control group (Table 10) show that students who obtained the best grades in the questions about the block "Politics in Latin America" in the final exam and in the final grades of the class tended to perform better in terms of self-efficacy. That is to say that the initial situation of the control group, with the significance of the soft skill studied in relation with the hard skills, was maintained after the experiment. The students' perception of their self-efficacy did not worsen, and they maintained the positive relationship with the academic performance (grades). Along these lines, it could be affirmed that the type of teaching format does not seem to influence the development of self-efficacy, which remains the same before and after the experiment in the control group.

In terms of critical thinking, the results of the PCs in the experimental group before the experiment do not show any significant correlation between this soft skill and any of the measurements of the hard skills (Table 4). Nevertheless, after the experiment, the results of the PCs show that there was a correlation between students who performed best in the final grade of the class and students who developed critical thinking the most (Table 8). While many studies generally detected the positive effects of active learning formats, including the FC, on students' critical thinking, only one study observed a consistency in terms of improving both grades and

**Table 13. Canonical correlation Set 2.**

| Variable | 1 | 2 | 3 |
|---|---|---|---|
| ContSE2 | .785 | .092 | .382 |
| ContTW2 | .440 | .573 | .145 |
| ContLP2 | .601 | -.217 | -.755 |
| ContCT2 | .361 | -.870 | .258 |

**Table 14. Variance explained canonical correlation.**

| Canonical variable | Set 1 by itself | Set 1 by set 2 | Set 2 by itself | Set2 by set 1 |
|---|---|---|---|---|
| **1** | .434 | **.199** | .325 | **.149** |
| 2 | .384 | .045 | .285 | .033 |
| 3 | .181 | .003 | .201 | .003 |

critical thinking [42]. Even though the authors separately measured the impact of the FC on hard and soft skills, our study goes in a similar direction. After simultaneously analysing the correlation between hard skills and critical thinking, we observed that the students with the best grades tended to develop stronger critical thinking skills.

In terms of perception of learning, the results of the PCs (Table 8) show that, in the experimental group after the experiment, there was a correlation with the students' scores in the final survey. This correlation was not detected before the experiment (Table 4). After the experiment, we found a coherence between what students achieved in their final grades of the class

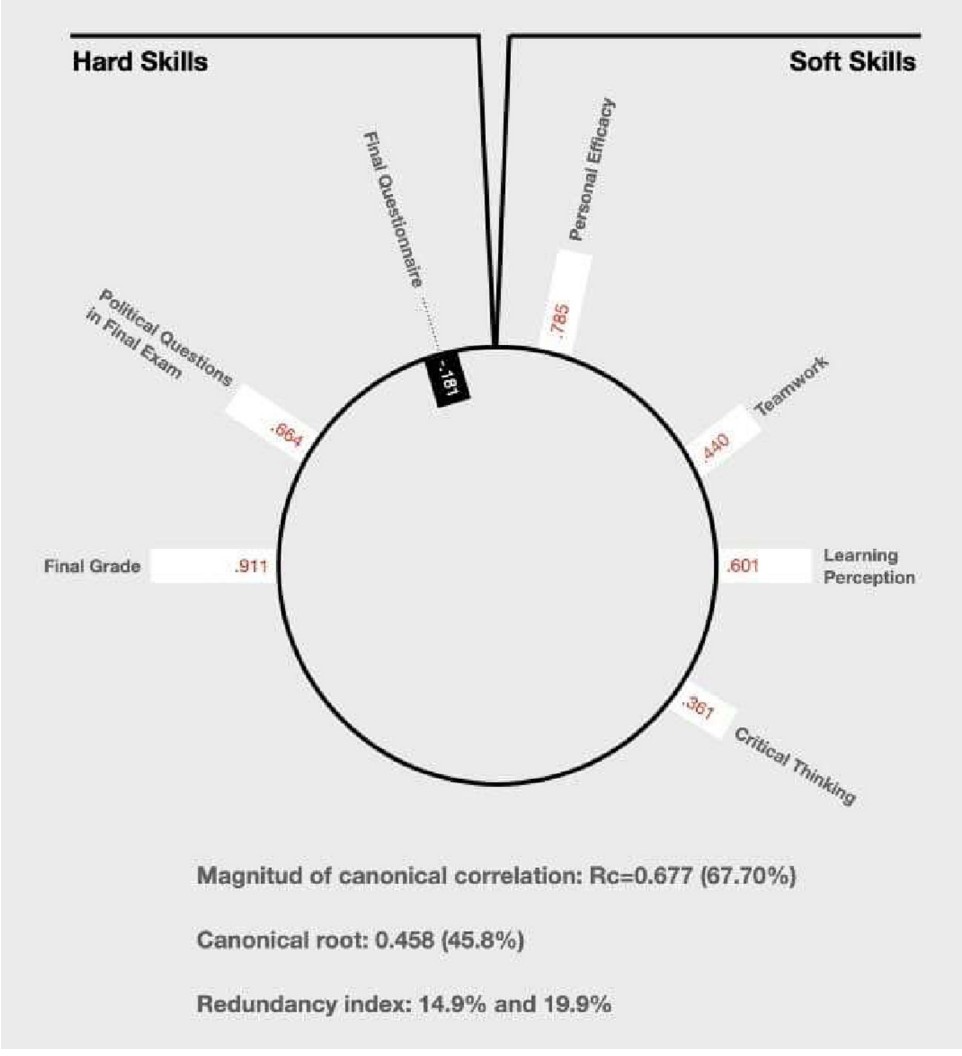

**Fig 1. Heliograph canonical correlation.**

and how they perceived their learning. This corroborates most previous studies [15, 17, 49, 54, 55], even though they separately analysed these two sets of variables. Unlike the results of other studies [19, 53], the students of our experiment did not associate active learning techniques, such as the FC, with an increased cognitive load.

It is interesting to observe that the only significant CCA between the sets of hard and soft skills is the one contained in Table 11, which shows the results of the control group after the experiment. We saw a convergence between hard and soft skills in this case, with a quite strong canonical root (45.8%) and two high redundancy indexes (14.9% and 19.9%). The only exception was the survey administered after the experiment, which had a negative sign, even though its canonical loading was not statistically significant. This could be the consequence of the fact that this survey was not a formal exam for which students properly prepared, but, rather, a control measure of the students' knowledge after the explanation of the contents and the performance of the class activities.

We found a positive sign in the other two indicators of the hard skills and all the soft skills. As was explained above, the control group was taught through a mixed teaching format, in which the traditional frontal face-to-face lectures were integrated with a variety of active learning techniques, as opposed to the experimental group, which was entirely taught through a flipped format. This could possibly suggest that semi-traditional teaching formats, based on face-to-face lectures integrated with active learning techniques, are more effective than fully active learning formats, such as FCs, on improving the knowledge of the contents and the development of soft skills. Yen et al. made a similar point in their comparison between face-to-face, online, and blended teaching formats [69]. This conclusion is also consistent with those scholars who observed that students tend to prefer "mixed class sessions" rather than "having all flipped-class sessions" [31].

While they do not deny the potentially large benefits of active learning formats, these authors invite us to be prudent, by considering, for example, the problems that these formats can pose, for both students and teachers, in terms of training and investments [69]. Other studies found that sometimes students are the ones who ask for prudence in the implementation of these techniques, especially in education systems that are strongly rooted to traditional teaching formats [9]. Finally, there seems to be an emerging consensus around the fact that FCs are more effective when they are implemented gradually, in a non-disruptive way, and, when they are integrated with other active learning strategies, such as peer-assessment [21, 20], self-evaluation mechanisms [13], cooperative learning techniques [14], games [42], or simulations [49]. Other authors similarly observed that FCs tend to be more useful, when they are enriched with other technological resources, such as Moodle [55], MOOCs [37] or other online formats [15].

In sum, our experiment showed that, even when they satisfactorily perform in terms of academic achievement, students can perform equally well only in some of the soft skills. Before the experiment, in the experimental group, teamwork correlated with two measurements of the hard skills. This correlation disappeared after the experiment. Students who performed best in teamwork were not the students who showed a better knowledge in any of the measurements of the hard skills. In terms of self-efficacy, the lack of convergence between this soft skill and some of the measurements of the hard skills held both before and after the experiment. This can indicate that the FC did not significantly influence the development of this soft skill.

Regardless of whether or not the FC or the traditional class integrated with other active learning methodologies, after the experiment, students performed better in both critical thinking and perception of learning. This happened with both teaching formats, which reinforces the idea that the type of active learning methodology was not necessarily the main factor to explain students' performance or the convergence between hard and soft skills.

Our results could also challenge one of the main expectations of the active learning literature: traditional lecturing is effective "in transmitting factual knowledge to students", but not equally effective for the development of students' "higher order cognitive skills" According to this view, active learning techniques are more effective for the development of such soft skills. [48]. Our experiment suggests that this opposition does not always correspond to the reality of teaching and learning. Integrating traditional and active teaching formats seems to be a better alternative than simply choosing one over the other.

## Conclusions, limitations and future research

These mixed results require further research on the capacity of active learning methodologies to improve students' hard and soft skills. The correlation between these two sets of variables seems to be less clear than what active learning studies would expect. While many studies tend to assume that active learning methodologies can equally improve achievement and soft skills, what our experiment has detected is a separate effect. Some soft skills converged with some of the measurements of the hard skills, while others did not. This can mean that we might need to use different active learning techniques, depending on what specific hard or soft skills we aim to develop or on what learning goals we aim to achieve [40]. More research is necessary on which specific students' skills can be improved by the FC or other active learning technique.

Moreover, more studies should be devoted to studying the possible drawbacks of the FC and other active learning techniques. One of the teachings of our study is that the FC cannot be taken as the panacea for all the learning necessities of students. More analyses are needed to find whether video-lectures can reduce the students' capacity to maintain attention [68] or to develop a collaborative environment. This is a particularly important issue in an epoch of pandemics that can oblige universities to move to online teaching.

Finally, more studies are necessary investigating the possible challenges to train teachers and students in the management of FC and other active learning formats. Mastering these formats can require a considerable economic investment in training students and teachers [1], which is something that universities need to carefully consider when planning to implement these formats.

Due to time and financial limitations, our study could not deal with these important aspects. We are also aware that our experiment took place in only one class over a single semester. Moreover, the University where the experiment took place strives to offer smaller classroom sizes in order to guarantee a more personalised education. Nevertheless, this can limit the possibility to detect robust patterns. For this reason, we will need to repeat the experiment by relying on larger samples of participants, on longer time spans, and by integrating the quantitative approach with qualitative techniques, such as interviews and focus groups with students. This would provide for a deeper understanding of students' perceptions, while facilitating the generation of new hypotheses and the achievement of more reliable information.

Despite these limitations, we also believe that this study provides an exploration on an important aspect of active learning, which is its capacity to enhance students' skills. It is our intention to expand our results by designing future projects based on a mixed method approach and on a combination of different qualitative and quantitative techniques to collect and analyse data.

## Supporting information

**S1 File.**
(XLSX)

## Author Contributions

**Investigation:** Andrea Betti, Aurora García Domonte.

**Project administration:** Pablo Biderbost.

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
