## [Decision Letter · Decision Letter 0]

22 Dec 2021

PONE-D-21-34007Can Active Learning Techniques Simultaneously Develop Students’ Hard and Soft Skills? Evidence from an International Relations ClassPLOS ONE

Dear Dr. Betti,

Thank you for submitting your manuscript to PLOS ONE. After careful consideration, we feel that it has merit but does not fully meet PLOS ONE’s publication criteria as it currently stands. Therefore, we invite you to submit a revised version of the manuscript that addresses the points raised during the review process.

We look forward to receiving your revised manuscript.

Kind regards,

Rong Zhu, Ph.D.

Academic Editor

PLOS ONE

Journal Requirements:

2. In the ethics statement in the Methods and online submission information, please clarify whether consent was written or verbal.  If verbal, please also specify: 1) whether the ethics committee approved the verbal consent procedure, 2) why written consent could not be obtained, and 3) how verbal consent was recorded. If your study included minors, state whether you obtained consent from parents or guardians. If the need for consent or parental consent was waived by the ethics committee, please include this information.

“The Pontifical University of Comillas, ICAI-ICADE (Madrid) financed the research for this article (Name of the research project: “Teaching Innovation in International Relations: A Comparative Study of the Flipped Classroom and the Traditional Classroom”).”

6. We note you have included a table to which you do not refer in the text of your manuscript. Please ensure that you refer to Table 1 and 2 in your text; if accepted, production will need this reference to link the reader to the Table.

Reviewers' comments:

Reviewer's Responses to Questions

**Comments to the Author**

1. Is the manuscript technically sound, and do the data support the conclusions?

Reviewer #1: Partly

2. Has the statistical analysis been performed appropriately and rigorously? 

Reviewer #1: Yes

3. Have the authors made all data underlying the findings in their manuscript fully available?

Reviewer #1: Yes

4. Is the manuscript presented in an intelligible fashion and written in standard English?

Reviewer #1: Yes

5. Review Comments to the Author

Reviewer #1: I really like this manuscript in terms of the questions that are being asked and the topics that are being covered. I have a number of empirical questions that I would like to see answered before publication.

• Are there differences in self-selecting into the survey between the two groups that you can speculate about or estimate? It could be that students in the treatment group notice that they are in the new, more interesting mode and are thus more willing to co-operate whereas students in the control group.

• To what extent was there contamination between the two groups? Groups knew about each other, groups could informally share information or teaching materials. What do you know about that? Did you ask about that in the survey? This spill-over could explain the smallish difference.

• Why was the experiment not pre-registered?

• The allocation of subjects into groups follows alphabetical order, an allocation mechanisms regularly employed by the university. This means that students in the treatment group were likely to have already co-studied with other peers in their group. How would such a mechanism possibly affect the results? You show that grades are on average the same in the two groups, but what if the control group individuals have already collaborated elsewhere before and thus have a super high level of soft skills?

• To what extent was the traditional format still traditional? It may be that the teachers who are so active to scientifically assess the teaching mode are already pretty great at teaching in the “traditional format”. Thus, the control group may have had too high competence levels because there were only good teachers around. How can you separate mode effects from teacher effects?

• How many latent dimensions of skills are there? I got confused by the plethora of variables.

• I have difficulties with understanding the need for the rather complex statistical analysis. In my view, this is a simple experimental design of Change_SoftSkill_n=a+b*TeachingMode for n soft skill variables. Simple tests with adjustment for multiple testing should do the trick in my book. Why do you go down the path of canonical correlations? It looks a bit like cracking nuts with sledgehammers.

• Can you replicate the study to get a higher number of N? Currently, you may have a problem of low statistical power.

Miscellaneous

We published a study in German where we demonstrate that FC improve the exam results in political methodology in exam questions for which students have to no apply knowledge rather than just reproduce: Goerres, Achim/Kärger, Caroline/Lambach, Daniel (2015): Aktives Lernen in der Massenveranstaltung: Flipped-Classroom-Lehre als Alternative zur klassischen Vorlesung in der Politikwissenschaft, Zeitschrift für Politikwissenschaft, 25/1: 135-52. (https://papers.ssrn.com/sol3/papers.cfm?abstract_id=2626020 )

6. PLOS authors have the option to publish the peer review history of their article (what does this mean?). If published, this will include your full peer review and any attached files.

Reviewer #1: **Yes: **Achim Goerres

---

## [Author Response · Author response to Decision Letter 0]

2 Feb 2022

RESPONSE TO REVIEWERS

To the attention of the academic editor and reviewers:

Thank you for the opportunity to submit a revised version of our paper. 

1) We submitted two copies of our paper. One is a marked-up copy of our manuscript that highlights changes made to the original version. We uploaded this as a separate file labelled “Revised Manuscript with Track Changes.” The other one is an unmarked version of our revised paper. We uploaded this as a separate file labelled 'Manuscript'.

2) Since the funders of our research project had no role in study design, data collection, analysis, decision to publish, or preparation of the manuscript, we added the following sentence to the financial disclosure, as suggested by the editor: “The funders had no role in study design, data collection and analysis, decision to publish, or preparation of the manuscript.”

3) We uploaded our study’s underlying data set in a separate file called “Supporting Information file.” We updated the cover letter to indicate that we uploaded the data set in a separate file. We also added this information at the beginning of the “Revised Manuscript with track changes” (highlighted in red), and at the beginning of the “Manuscript.”

4) We double-checked all manuscripts to make sure that they meet PLOS ONE's style requirements, including those for file naming.

5) In the section of the manuscript called “Ethics Statement” (pp.10-11), we clarified that students voluntarily accepted to participate in the research. Moreover, we explained that their consent and acceptance to participate in the research was electronic. This means that they provided their personal information in Moodle. Finally, we added that there were no minors involved in the project.

6) Financial disclosure: see point 2 of this memo.

7) Data set: see point 3 of this memo.

8) We validated our ORCID ids in the Editorial Manager.

9) Since we had not referred to either Table 1 or Table 2 in the first submitted version of the manuscript, we added a reference to Table 1 on page 18, at the end of the first paragraph, and a reference to Table 2 at the end of the second paragraph.

10) To respond to the point raised by Reviewer 1 about the availability of our data set, see point 3 of this memo, where we explained that we uploaded our study’s underlying data set in a separate file called “Supporting Information file.”

11) We agree with Reviewer 1 that there can be differences in self-selecting into the survey between the two groups that we could speculate about. However, in the second paragraph of page 15, we explained that we delivered two ANOVA tests, which did not detect any statistically significant difference in the students’ academic achievement prior to our experiment (ANOVA IR GDA p= 0.847; ANOVA BA GDA p=0.716). We believe this means that the sample was homogenous in terms of academic achievement and that there was no statistically significant difference between the two groups. To double-check this point, we also compared our distribution (alphabetical order) with two other possible ways to divide students into groups, such as student identification number and a fully random distribution. We did not find any significant difference, with the two groups being homogenous. This was explained on page 15. We also agree with Reviewer 1 that it could be that students in the treatment group notice that they are in the new, more interesting mode, and are thus more willing to co-operate, as compared to students in the control group. Although this was not the case in our experiment (see point 12 of this memo), we believe that it is an important point that we will certainly consider in future experiments.

12) We thank Reviewer 1 for the point about the possibility of contamination between the two groups. It is an important point that needs to be seriously considered in these types of experiments, as this spill-over could explain the differences. Nevertheless, while we were doing the research for this project, we conducted two focus groups with a sample of students from each group. We observed that students of each group did not know about each other, and that there was no exchange of information between them on this point. The main reason for this lack of contamination is that materials in Moodle were only accessible for the members of each group. Members of each group could not access the materials of the other group. In this sense, we did not detect any contamination between the experimental and the control group. We believe that this is an important point, so we added a sentence about this in the third paragraph of page 10. 

13) Our experiment was not pre-registered. The main reason is that our funders did not require us to do it or to publish the preliminary results of the experiment in an open access working paper. However, we think this is an important suggestion that can open opportunities for interesting discussions and exchange of ideas with other researchers. For this reason, we will certainly consider it for future experiments.

14) As to the allocation of subjects into groups by following an alphabetical order, we did not use natural groups, which means groups freely formed by students. Rather, in the University where the experiment took place, and also in our class, we used artificial groups, which means groups created by the professor or by the University. This means that students usually work in different groups, depending on the different tasks and the different subjects. We added a sentence on this point in the second paragraph of page 10.

15) To avoid the differences between the mode effects and the teacher effects, a point raised by Reviewer 1, we mentioned on page 12 that the professor was the same for both the experimental and the control group. The reason for this is that we wanted to study only the differences between the two teaching methodologies (flipped vs. semi-traditional). However, this point made us think about the possibility to compare not only the different teaching methodologies, but also different professors. We could not do this for this paper, but we plan on conducting a new experiment in which we can study the students’ performance by following this logic:

One control group and one experimental group with the same professor for both groups.

Another control group and another experimental group with another professor for both groups.

16) As to the latent dimensions mentioned by Reviewer 1, we did not analyse latent dimensions for this study. What we did do was select the soft skills and the indicators to measure them on the basis of the literature (see pp. 6-10). Studying latent dimensions would have required the use of different statistical techniques. We will certainly consider doing this in a future experiment. 

17) As we explained on page 19-20, we used Canonical Correlation Analysis to detect the level of association between both sets of variables, hard and soft skills, for each group, according to a before-after logic. This is useful to maximise the correlations among lineal combinations of variables and to simultaneously predict the behaviour of multiple variables. We believe that this technique of multivariate statistics was adequate to achieve our research goals. 

18) As Reviewer 1 indicates, it is true that we would need to replicate the study to get a higher number of N. This would increase the statistical power of the experiment. We will take this into account for future studies, in which we will work with more data and with larger samples. So far, we have worked with the numbers that we could obtain by conducting the experiment in a single class in a University that usually avoids creating too large of classrooms. This is an important limitation of our study which we further clarified on page 35.

19) We really thank Reviewer 1 for the suggestion about the article that shows how the Flipped Classroom can improve the exam results in questions for which students have to apply rather than just reproduce knowledge. Since we believe that it is a very interesting study, we added it to the literature review (p. 5, note 24) and to the list of sources (p. 38).

---

## [Decision Letter · Decision Letter 1]

2 Mar 2022

Can Active Learning Techniques Simultaneously Develop Students’ Hard and Soft Skills? Evidence from an International Relations Class

PONE-D-21-34007R1

Dear Dr. Betti,

We’re pleased to inform you that your manuscript has been judged scientifically suitable for publication and will be formally accepted for publication once it meets all outstanding technical requirements.

Kind regards,

Rong Zhu, Ph.D.

Academic Editor

PLOS ONE

Additional Editor Comments (optional):

Reviewers' comments:

Reviewer's Responses to Questions

**Comments to the Author**

1. If the authors have adequately addressed your comments raised in a previous round of review and you feel that this manuscript is now acceptable for publication, you may indicate that here to bypass the “Comments to the Author” section, enter your conflict of interest statement in the “Confidential to Editor” section, and submit your "Accept" recommendation.

Reviewer #1: (No Response)

2. Is the manuscript technically sound, and do the data support the conclusions?

Reviewer #1: (No Response)

3. Has the statistical analysis been performed appropriately and rigorously? 

Reviewer #1: Yes

4. Have the authors made all data underlying the findings in their manuscript fully available?

Reviewer #1: Yes

5. Is the manuscript presented in an intelligible fashion and written in standard English?

Reviewer #1: Yes

6. Review Comments to the Author

Reviewer #1: Dear authors,

thank you very much for your clear revisions. I think that such a project as yours, a field experimental study of the efficacy of teaching methods, is examplary for the improvement of teaching. Future projects, if you have planned any, should have pre-registered hypotheses and analysis plans.

best

reviewer 1

7. PLOS authors have the option to publish the peer review history of their article (what does this mean?). If published, this will include your full peer review and any attached files.

Reviewer #1: **Yes: **Achim Goerres

---

## [Editor Report · Acceptance letter]

8 Mar 2022

PONE-D-21-34007R1 

Can Active Learning Techniques Simultaneously Develop Students’ Hard and Soft Skills? Evidence from an International Relations Class 

Dear Dr. Betti:

I'm pleased to inform you that your manuscript has been deemed suitable for publication in PLOS ONE. Congratulations! Your manuscript is now with our production department. 

Kind regards, 

on behalf of

Dr. Rong Zhu 

Academic Editor

PLOS ONE